# Enhancing Radiotherapy Sensitivity in Prostate Cancer with Lentinan-Functionalized Selenium Nanoparticles: Mechanistic Insights and Therapeutic Potential

**DOI:** 10.3390/pharmaceutics16091230

**Published:** 2024-09-21

**Authors:** Yani Zou, Helin Xu, Xiu Wu, Xuesong Liu, Jianfu Zhao

**Affiliations:** 1Department of Oncology of the First Affiliated Hospital, Jinan University, Guangzhou 510660, China; zouyn9981@163.com (Y.Z.); liuxuesong@stu2021.jnu.edu.cn (X.L.); 2Research Center of Cancer Diagnosis and Therapy, Jinan University, Guangzhou 510632, China; 3Tumor Radiotherapy Center, Fuyang People’s Hospital, Fuyang 236012, China; 4Department of Emergency Surgery, Fuyang People’s Hospital, Fuyang 236012, China; 13956818795@163.com; 5Department of Clinical Pathology, Linyi Maternal and Child Healthcare Hospital, Linyi 276016, China; luckyone202405@163.com

**Keywords:** LET-SeNPs, prostate cancer, radiosensitization, ROS, apoptosis

## Abstract

Radiation therapy is a cornerstone of prostate cancer (PCa) treatment. However, its limited tumor sensitivity and severe side effects restrict its clinical utility. Lentinan-functionalized selenium nanoparticles (LET-SeNPs) have shown promise in enhancing radiotherapy sensitivity and exhibiting antitumor activity. In this study, we investigated the radiotherapy sensitization mechanism of LET-SeNPs in PCa. Our results demonstrate that the combination of LET-SeNPs and X-ray therapy (4 Gy) significantly inhibited the growth and colony formation of PCa cells by inducing apoptosis, surpassing the effects of individual treatments. This combined approach modulated DNA damage through the p53, MAPK (mitogen-activated protein kinase), and AKT pathways. Furthermore, LET-SeNPs increased PC3 cell sensitivity to X-ray-induced apoptosis by downregulating TrxR (Thioredoxin reductase) expression and inducing reactive oxygen species (ROS) overproduction, thereby activating mitochondria-mediated apoptosis signaling pathways. Additionally, LET-SeNPs regulated PARP (poly (ADP-ribose) polymerase) to prevent DNA damage repair. In vivo studies confirmed that the combination treatment inhibited PCa growth by synergistically activating the p53 pathway to induce cell apoptosis. These findings highlight LET-SeNPs’ potential as a radiotherapy sensitizer and suggest that combining LET-SeNPs with X-ray therapy could be a promising strategy for clinical application, leveraging selenium-modified nanoparticles’ antitumor effects.

## 1. Introduction

Prostate cancer (PCa) ranks as the most prevalent male-related malignant tumor in 112 countries globally [1] and stands as the second-most common male malignancy following lung cancer [2,3]. The current standard of care for advanced prostate cancer involves a combination of conventional chemotherapy, radiotherapy, and targeted anti-PD-L1 monoclonal antibodies [4]. Notably, prostate cancer harbors specific tumor antigens like PAP, PSCA, and PSMA, which are pivotal targets for immunotherapy [5,6,7] and CAR T therapy [8,9]. Radiation therapy plays a crucial role in the medical management of prostate cancer by effectively impeding tumor spread and metastasis. High-dose radiation therapy, although recommended for maximizing survival benefits, can lead to significant adverse effects on the urinary and digestive systems [10,11,12]. Accurate control of the treatment site and radiation dosage is imperative [13,14,15], underscoring the need for effective radiosensitizers to enhance radiotherapy outcomes [16,17]. Consequently, a series of drugs known as radiosensitizers have been developed to enhance radiotherapy to PCa [18].

Radiotherapy sensitizers enhance the tumor response to radiation through various physical mechanisms. One critical aspect is the improvement of hypoxia. Rapid tumor growth often leads to a hypoxic environment, significantly reducing the efficacy of radiotherapy; under hypoxic conditions, the radiation dose required to kill tumor cells is 2–3 times higher than that under aerobic conditions. Therefore, alleviating hypoxia can improve treatment outcomes [19]. Another important mechanism involves the regulation of the cell cycle. Tumor cell sensitivity to radiotherapy is closely linked to the cell cycle; G2-M phase cells are most susceptible to radiation, whereas G1 phase cells exhibit lower sensitivity. If G2-M phase cells undergo cycle arrest, the overall sensitivity to radiotherapy increases [20]. Lastly, the core of radiotherapy lies in inducing DNA damage, primarily through single-strand and double-strand breaks, as well as base damage [21]. Double-strand breaks are the predominant form of DNA damage, and nanodrugs can inhibit DNA repair mechanisms, thereby enhancing the overall effectiveness of radiotherapy.

Selenium, a vital micronutrient for humans, animals, and plants [22,23], has garnered attention for its potential in cancer prevention and treatment [24,25]. Selenium nanoparticles with functionalized surfaces have exhibited promising anticancer properties and sensitization to radiotherapy [16,26,27]. For instance, the nanosystem SeD@MSNs-FA, comprising Selenodiazole (SeD) and folic acid (FA), has shown efficacy in inhibiting cervical cancer cells by inducing S phase arrest [28]. Similarly, the combination of polyethylene glycol nano-selenium (PEG-SeNPs) with X-ray therapy has demonstrated growth inhibition in lung cancer cells by inducing G2-M phase arrest and elevating reactive oxygen species levels [29].

Studies have highlighted the importance of selenium’s specific form in inducing cell cycle arrest and apoptosis in tumor cells [29]. LET-SeNPs have shown superior antitumor activity compared to other selenium compounds, with significant selectivity between tumor and normal cells [30]. The combination of LET-SeNPs with X-ray therapy has been found to enhance reactive oxygen species build-up in cancer cells, leading to DNA damage mediated by mitochondria [30]. LET-SeNPs serve as potent radiosensitizers, enhancing radiotherapy efficacy while maintaining low toxicity levels. The immunomodulatory and antitumor activities of mushroom polysaccharides have been extensively studied, particularly in the context of their role as natural adjuvants for tumor chemotherapy in Asia [31,32]. The unique chemical structure of mushroom polysaccharides, characterized by a high number of hydroxyl groups, facilitates their strong physical adsorption onto selenium nanoparticles (SeNPs), thereby preventing their accumulation and precipitation [33]. Previous research efforts have focused on the construction and evaluation of mushroom polysaccharide-decorated SeNPs, which have shown enhanced uptake by cancer cells in cancer therapy [34].

This study investigates the physical properties of functionalized selenium nanoparticles (LET-SeNPs) and their mechanisms of action in conjunction with X-ray therapy against prostate cancer PC3 cells. In addition to these in vitro analyses, we established both unilateral and bilateral nude mouse models to evaluate the in vivo antitumor activity and toxicity of LET-SeNPs. Utilizing these animal models allows for a comprehensive assessment of the therapeutic efficacy and safety profile of LET-SeNPs within a living organism, providing valuable insights into their potential clinical applications. The evaluation of antitumor effects in vivo complements our in vitro studies, reinforcing the translational potential of LET-SeNPs as effective radiosensitizers in prostate cancer treatment. Collectively, the findings from this research significantly enhance our understanding of LET-SeNPs, supporting their further development as a promising strategy to improve radiotherapy outcomes for prostate cancer patients.

## 2. Materials and Methods

### 2.1. Materials and Chemistry

DCF probe (Sigma-35845-1G) was purchased from Sigma-Aldrich (Guangzhou, China), JC-1 (T4069-5MG) was purchased from Sigma-Aldrich (Shenzhen, China). Invitrogen (Guangzhou, China) provided the fetal bovine serum (FBS), and propidium iodide (PI) was purchased from Sigma-Aldrich. LET (L413225-25mg) was purchased from Aladdin (Guangzhou, China).

### 2.2. Preparation and Characterization of Mushroom Polysaccharide-Decorated SeNPs (LET-SeNPs)

The preparation and characterization of LET-SeNPs were conducted following a previously reported procedure [35]. In brief, a 50 mM sodium selenite (Na_2_SeO_3_) solution (1 mL) was mixed with a 10 mg/mL LET solution (2 mL) under continuous, low-speed stirring. Milli-Q water was then added to reach a final volume of 9 mL. Subsequently, a 200 mM vitamin C (Vc) solution (1 mL) was carefully added dropwise to the mixture. The solution was magnetically stirred at room temperature for 12 h and then dialyzed in Milli-Q water for 24 h to eliminate excess Na_2_SeO_3_ and Vc. The resulting nanosystem, referred to as LET-SeNPs, was stored at 4 °C. Digestion of the sample was carried out by mixing hydrochloric acid (HClO_4_) and nitric acid (HNO_3_) in a 1:3 volume ratio. The concentration of freshly prepared LET-SeNPs was determined to be 3.33 mmol/L using a dosimeter. The morphology of the LET-SeNPs was visualized through high-resolution transmission electron microscopy (TEM) imaging and the surface charge was assessed using a zeta potential analyzer (Zetasizer Nano ZSE, Malvern Instruments Limited, London, UK). Additionally, the stability of LET-SeNPs was investigated in PBS, FBS, and DMEM complete media.

### 2.3. Cell Culture and Cytotoxicity Assay

The PC3 cell line obtained from ATCC was cultured in DMEM medium supplemented with 10% fetal bovine serum and 1% anti-(penicillin–streptomycin) under 5% CO_2_ conditions. Cell viability after treatment with various concentrations of LET-SeNPs combined with X-ray radiation (2, 4, 8, 16, and 32 Gy) was assessed using the MTT assay [36].

### 2.4. Clonogenic Assays

PC3 cells were treated with different concentrations of LET-SeNPs (1, 2, and 4 μM) for 4 h followed by X-ray irradiation (4 Gy). The cells were then incubated for 7–14 days, fixed with 4% paraformaldehyde, stained with 0.5% crystal violet, and evaluated for clone viability.

### 2.5. Flow Cytometric Analysis

PC3 cells were exposed to varying levels of LET-SeNPs (1, 2, and 4 μM) and X-ray radiation (4 Gy). Subsequent treatment with propidium iodide (PI) or Annexin V/PI allowed the evaluation of the combined impact of LET-SeNPs and X-ray radiation using a Beckman CytoFLEX S flow cytometer (Brea, CA, USA).

### 2.6. Mitochondrial Membrane Potential (ΔΨm) Measurement

Following the same cell processing method as above, cells were treated with JC-1 solution after 72 h. Flow cytometry analysis was performed to measure ΔΨm.

### 2.7. Detection of Levels of Reactive Oxygen Species (ROS)

The effect of LET-SeNPs combined with X-ray on ROS was evaluated using the DCFH-DA fluorescent probe by a microplate reader(Agilent BioTek Cytation 5, Santa Clara, CA, USA) [37]. PC3 cells were plated in 96-well plates at a density of 2 × 10^5^ cell/mL and left to attach for 24 h. Subsequently, different concentrations of LET-SeNPs (1, 2 and 4 μM) were added to the cells. DCFH-DA-DMEM (20 μM) was then introduced for staining and incubated for 30 min, X-ray (4 Gy) radiation was immediately administered. The fluorescence levels were assessed by a microplate reader set at 488 nm for excitation and 525 nm for emission, with readings recorded over a 2 h period. Furthermore, a fluorescence microscope was utilized to observe DCF fluorescence levels.

### 2.8. Caspase-3 Viability

The PC3 cells were incubated in dishes for 24 h and then treated with LET-SeNPs (1, 2 and 4 μM), and X-ray radiation (4 Gy) for 4 h. Then, RIPA was introduced, and an equal amount of Ac-DEVD-AMC (caspase-3/7) was added [38]. Caspase function was subsequently identified by a microplate reader that utilized a fluorescence excitation wavelength of 365 nm and an emission wavelength of 450 nm. The absorbance values were recorded, and the relative activity of each control group was calculated.

### 2.9. Western Blot Analysis

Firstly, the samples were prepared according to groups. The protein concentration was measured after lysis and the protein samples were prepared by adding RIPA and SDS buffer, boiled at 100 °C for 10 min, and stored at −20 °C. The gel and running buffer were prepared for electrophoresis, which was run at 150 V for 100 min, and marked with 310 KDMARK. The transfer buffer was then prepared, the gel transfer sandwich assembled in the following arrangement, sponge–filter paper–gel membrane–filter paper–sponge, transferred at 240 mA for 110 min, incubated with the corresponding antibodies after cutting the membrane, incubated overnight in a 4 °C refrigerator, washed three times with TBST, incubated with secondary antibodies for 2 h, washed three times with TBST, added to the exposure solution, and exposed [39].

### 2.10. Animal Model

Male BALB/C nude mice (22–28 days old) were obtained from Zhuhai BST Bioscience and housed at the Jinan University Laboratory in compliance with the ARRIVE guidelines. Experimental procedures were carried out following approved protocols (Approval No: IACUC-20240619-09) by the Experimental Animal Ethics Committee at Jinan University (Guangzhou, China). PC3 cells at a concentration of 2 × 10^7^ cells/mL were injected into the unilateral armpit and then divided into 4 groups (n = 5), randomly. The mice in the experimental groups received either an intravenous injection of LET-SeNPs at a dose of 2 mg/kg or were exposed to X-ray (4 Gy) radiation twice weekly, or a combination of both treatments. Only PBS was administered to the control group. Tumor volume was measured over time, and after 21 days of treatment, the mice were euthanized. Blood was collected, and the tumor as well as major organs including the heart, liver, spleen, lung, and kidney were dissected for further analysis [16].

Next, 5 nude mice were randomly selected, with the same standards as the aforementioned unilateral tumor mice. To create a bilateral tumor model, 200 μL of PC3 cell suspension (100 × 10^4^/100 μL) was injected into the left and right armpits of the mice. At around 20 mm^3^, LET-SeNPs were given at 2 mg/kg through tail vein injection on days 3, 6, 9, 12, 15, 18, and 21. Within 4 h of administration, the right armpit tumor areas of the mice were irradiated with X-rays at a single dose of 4 Gy, totaling 28 Gy, while shielding the opposite side with a 2 cm lead plate for radiation protection. All radiation experiments are conducted using an Elekta Precise linear accelerator, which operates at 6 MeV energy, provides a penetration depth of 5 cm, and delivers 100 monitor units (MU) in a hospital setting.

### 2.11. Statistical Analysis

Each experiment was repeated at least three times, and the results are presented as mean ± standard deviation. Statistical significance was determined using a *t*-test or one-way ANOVA with GraphPad Prism version 9.2.

## 3. Results

### 3.1. Morphology and Stability of LET-SeNPs

The functional modification of selenium nanoparticles (SeNPs) was achieved through a simple method involving mushroom polysaccharides, resulting in the formation of compact and stable spherical nanostructured materials, as depicted in the preparation flowchart in Figure 1A. The excellent dispersibility of LET-SeNPs with particle sizes around 100 nm is evident in Figure 1B,C. In this study, the prepared nano sample of LET-SeNPs was incubated in various proportions with PBS, FBS, and DMEM culture medium. The changes in particle size of the LET-SeNPs in different solvents over a 30-day period were monitored using a Nano-ZS nanoparticle size analyzer (Malvern Instruments Limited, London, UK) to assess the stability of the LET-SeNPs nano system. The results demonstrate that LET-SeNPs can maintain stability in PBS for a duration of 30 days, with minimal alterations in particle size (Figure 1D). In FBS and DMEM solvents, particle stability can be sustained for 5–7 days before a gradual increase in size occurs, ensuring the smooth progression of subsequent experimental studies. The high stability of LET-SeNPs under physiological conditions provides a solid foundation for their broad application in the medical and healthcare fields. Furthermore, the average zeta potential of LET-SeNPs was determined to be −9.68 mV, indicating a slight negative charge (Figure 1E).

### 3.2. Radiosensitization of LET-SeNPs Combined with X-ray

The cytotoxicity of selenium-containing compounds combined with X-ray radiotherapy on PC3 cells was investigated using the MTT method [36]. LET-SeNPs exhibited a significant inhibitory effect on cell proliferation in PC3 cells after 72 h of treatment, with the inhibitory effect increasing with drug concentration (Figure 1F).

The IC_50_ value provides an intuitive reflection of drug cytotoxicity, with cytotoxicity being inversely correlated with its value. The IC_50_ of LET-SeNPs is 81.22 μM when administered alone. When combined with X-ray 2 Gy, the IC_50_ decreased to 16.64 μM and further dropped to 5.64 μM with X-ray 4 Gy (Figure 1G).

Using the isobologram method, synergistic effects of LET-SeNPs combined with X-ray radiation at the 2 Gy and 4 Gy doses were observed, with corresponding LET-SeNP concentrations of 16.6 μM and 5.6 μM, respectively (Figure 1H). Subsequent experiments will involve X-ray irradiation (4 Gy) to investigate the radiosensitizing effect of LET-SeNPs at different concentrations (1, 2, and 4 μM).

Subsequently, the impact of pure X-ray irradiation on PC3 cytotoxicity was explored. Pure X-ray irradiation showed a dose-dependent decrease from 2 Gy to 32 Gy in cell viability, with an IC_50_ of 26 Gy for PC3 cells (Figure 1I).

### 3.3. LET-SeNPs Combined with X-ray Effectively Inhibits PC3 Cell Growth

A clonogenic assay was conducted to confirm the radiosensitization impact of LET-SeNPs and assess their ability to inhibit cell growth. The radiotherapy inhibition rates of LET-SeNPs exhibited a positive correlation with drug concentrations (Figure 2A,B). Interestingly, it was observed that LET-SeNPs could slightly promote the growth of PC3 cells. Specifically, at 2 μM, the clonal inhibition rate of LET-SeNPs decreased from 87.8% to 74.19% when combined with X-ray (4 Gy), and at 4 μM, it decreased from 11.07% to 7.08%. When irradiated alone, the clonal inhibition rate was 68.77%. Consistently, the findings from this clonogenic study showed that LET-SeNPs, when used in conjunction with X-ray (4 Gy) radiotherapy, were more successful in suppressing the proliferation of PC3 cells compared to using the drug alone or the same dose of radiotherapy in isolation. This highlights the potential of LET-SeNPs as a radiosensitizer in enhancing the therapeutic outcomes of radiotherapy in cancer treatment.

### 3.4. Induction of PC3 Cell Apoptosis

DNA damage is a critical factor mediating cell cycle arrest and apoptosis in the context of anti-proliferative interventions such as drug treatment and radiotherapy [40]. Apoptosis serves as the primary mechanism through which PC3 cell proliferation is impeded. Therefore, flow cytometry was employed to assess the cell cycle of PC3 cells following drug treatment and X-ray radiotherapy. The sub-G1 apoptosis peak in the X-ray (4 Gy) radiotherapy group exhibited a substantial increase from 0.16% to 16.65% relative to the control group. Notably, the sub-G1 apoptotic peak increased in a concentration-dependent escalation in response to LET-SeNPs as standalone treatment. Furthermore, the addition of X-ray (4 Gy) radiotherapy further potentiated the sub-G1 apoptotic peak. Specifically, at a concentration of 4 μM LET-SeNPs, the proportion of sub-G1 cells in the combined X-ray (4 Gy) radiotherapy group escalated from 14.71% to 28.36% compared to LET-SeNP monotherapy (Figure 3A,B).

It was observed that the radiotherapy group exhibited significantly higher proportions of both early and late apoptosis compared to the control group, with late apoptosis predominating over early apoptosis. The incorporation of LET-SeNPs into radiotherapy resulted in a concentration-dependent augmentation in the late apoptosis rate, escalating from 0.8% to 18.92% (Figure 3C,D). These findings underscore the efficacy of the LET-SeNPs and X-ray radiation (4 Gy) combination in suppressing PC3 cell growth through the facilitation of apoptotic processes.

### 3.5. Inducing the Decrease in the Mitochondrial Membrane Potential

Mitochondria play a crucial role in intracellular signaling pathways, making the monitoring of changes in mitochondrial membrane potential a key aspect of early apoptosis detection. Mitochondrial membrane potential reduction can trigger a cascade of events, including caspase protease family activation and ATP synthesis inhibition. Alterations in mitochondrial transmembrane potential are pivotal in initiating the apoptotic cascade. Therefore, we investigated changes in PC3 cell mitochondrial membrane potential following treatment with LET-SeNPs in conjunction with X-ray (4 Gy) radiotherapy to evaluate apoptosis.

It is shown that treating PC3 cells with varying amounts of LET-SeNPs (1, 2, and 4 μM) resulted in a reduction in the mitochondrial membrane potential. The percentage of cells showing a decrease in mitochondrial membrane potential were 6.56%, 15%, and 25.1% for each concentration, respectively. Exposure to X-ray radiation at a dose of 4 Gy led to a rise in the percentage of cells exhibiting reduced mitochondrial membrane potential from 3.95% to 18.8% when compared to the untreated group (Figure 4A). Furthermore, the combination treatment resulted in a notable escalation in the percentage of cells with reduced mitochondrial membrane potential, rising from 6.56%, 15%, and 25.1% to 26.2%, 30.8%, and 55.8% compared to the LET-SeNP treatment alone.

Higher drug concentrations of LET-SeNPs resulted in decreased red fluorescence and increased green fluorescence in PC-3 cells compared to the control group (Figure 4B and Appendix A). The findings suggested that LET-SeNPs and X-ray (4 Gy) radiation triggered cell death in PC3 cells through the activation of internal signaling pathways in the mitochondria. Moreover, the combination of LET-SeNPs with X-ray radiation (4 Gy) can amplify the reduction in mitochondrial membrane potential and facilitate cellular apoptosis.

### 3.6. Activation of ROS-Mediated Signaling Pathways

The intricate interplay between mitochondria and free radicals in mammalian cells underscores the significance of TrxR, an enzyme vital for maintaining cellular redox balance and signaling cell death. Inhibition of TrxR has been linked to radiation-induced DNA damage and the promotion of apoptosis through ROS-mediated DNA damage pathways [28,41].

Reactive oxygen species (ROS) serve as crucial regulators of cell death, influencing cellular metabolism and apoptosis signaling [42,43]. After administering LET-SeNPs in combination with X-ray (4 Gy) radiotherapy, we assessed ROS levels in PC3 cells. The absorbance of the DCF fluorescent probe was continuously monitored in the following 2 h. It depicts a significant increase in ROS levels within the initial 20 min, followed by a gradual rise after a minor decline when LET-SeNPs were combined with X-ray therapy. When LET-SeNPs were administered independently, there was a gradual increase in ROS levels over a period of 40 min, with the strength of the response correlating with the concentration of the nanoparticles (Figure 4C). Nevertheless, the enhancement was notable when paired with X-ray (4 Gy) radiation therapy. It shows that the fluorescence intensity was significantly higher in the LET-SeNP group when combined with X-ray radiotherapy compared to the other groups, and this increase was dependent on the concentration (Figure 4D and Appendix A). These fluorescence imaging results align with the concept that LET-SeNPs combined with X-ray radiotherapy can induce a larger amount of ROS in PC3 cells compared to other groups. These results indicate that the synergy between LET-SeNPs and X-ray radiotherapy induces higher ROS levels in PC3 cells, ultimately promoting cell death.

### 3.7. Activation of Downstream Signaling Pathways

Radiation therapy can induce DNA damage by activating various downstream signaling pathways [44], such as p53. The process also involves mitogen-activated protein kinases (MAPKs) such as p38, ERK, JNK, and AKT [45]. Therefore, we conducted an investigation into the expression and phosphorylation of proteins in these signaling pathways. As depicted in Figure 5A, the combination of LET-SeNPs with X-ray resulted in an increased expression of phosphorylated CHK-2 and phosphorylated p53, while the total p53 level remained unchanged, compared to radiotherapy alone. Additionally, as shown in Figure 5B, the combination treatment up-regulated phosphorylated p38, down-regulated antiapoptotic EKR, and did not affect the total JNK. Phosphorylated AKT, which is associated with cell proliferation [46], was suppressed. These pathways collectively up-regulated caspase-3, ultimately leading to DNA damage and apoptosis.

It illustrates the combined effect of LET-SeNPs and X-rays on reducing TrxR expression in PC3 cells. The diminished TrxR expression disrupts intracellular redox equilibrium, ultimately triggering apoptosis. This highlights the potent radiosensitization effects of LET-SeNPs (Figure 5B).

Rapid DNA damage and repair in tumor cells represent a protective mechanism following radiotherapy. Moreover, the down-regulation of poly (ADP-ribose) polymerase (PARP) expression in the combination group compared to X-ray radiotherapy alone inhibits the self-repair capability of PC3 cells [28], underscoring the potential therapeutic efficacy of the combined LET-SeNPs and X-ray treatment in suppressing tumor cell recovery mechanisms post-radiation exposure (Figure 5B).

Caspase-3 levels, a key player in apoptosis, were quantified using fluorescence analysis [47,48]. It shows that using LET-SeNPs by itself led to a small rise in caspase-3 levels inside the cells when compared to the control group. When X-ray was added, the caspase-3 levels in PC3 cells increased significantly in a dose-dependent fashion (Figure 5C). Overall, the LET-SeNPs and X-ray (4 Gy) radiation combination increased the caspase-3 levels in PC3 cells, leading to apoptosis activation and programmed cell death.

### 3.8. In Vivo Radiosensitization

In vivo, the cancer radiosensitization effects of LET-SeNPs were studied using PC3 tumor-bearing nude mice. PC3 cells were injected into the nude mice and divided into four groups, receiving LET-SeNPs at a dose of 2 mg/kg through the veins or exposed to X-ray radiation (4 Gy) twice weekly (28 Gy/7 F), or a combination of the two approaches. The tumor growth curves demonstrated the inhibitory effects of LET-SeNPs combined with X-ray. Under single X-ray radiotherapy and single LET-SeNP treatment, tumor growth was slowed. However, the combination treatment group exhibited a slower growth rate and smaller tumor volume. Following 28 days of therapy, the mean tumor sizes for the control, LET-SeNP, and X-ray groups were 1.36 cm^3^, 1.11 cm^3^, and 0.97 cm^3^, respectively, whereas the average tumor volume in the LET-SeNPs combined with X-ray radiotherapy group was notably reduced to 0.32 cm^3^. Various therapeutic approaches did not result in a notable change in the mice’s body weight across all groups, with average weights consistently ranging from 20 to 25 g (Figure 6A–D). This brief period of weight stability further demonstrates the efficacy and safety of the LET-SeNPs–X-ray radiotherapy combination. These findings indicate that the use of LET-SeNPs in conjunction with X-ray radiation therapy led to improved suppression of tumor growth in mice.

The LET-SeNPs combined with X-ray group showed the least nuclear and division abnormalities, along with the highest level of tissue differentiation based on H&E staining (Figure 6E). Immunohistochemical p-p53 staining showed that p-p53 was predominantly induced by the combination treatment. In conclusion, the use of LET-SeNPs in combination with X-ray radiation therapy demonstrated enhanced suppression of tumor growth in PC3 tumor-bearing nude mice, emphasizing the potential therapeutic benefits of this synergistic approach in inhibiting tumor progression in vivo.

The toxicity assessment of LET-SeNPs combined with X-ray therapy in nude mice included the examination of major organs and blood biochemical indexes using H&E staining. The organs assessed for potential toxicity included the heart, liver, spleen, lung, and kidney, which are critical for overall physiological function. Additionally, blood biochemical indexes such as AST (aspartate aminotransferase) and UA (uric acid) were analyzed to evaluate the systemic effects of the combined treatment. The results of the toxicity assessment revealed that the combination of LET-SeNPs and X-ray therapy exhibited strong antitumor efficacy while demonstrating minimal toxicity towards the mice. H&E staining of the major organs did not show significant signs of damage or abnormalities, indicating that the treatment did not cause adverse effects on the heart, liver, spleen, lung, or kidney (Figure 7A).

Furthermore, the analysis of blood biochemical indexes, including AST and UA levels, did not indicate any significant alterations that would suggest systemic toxicity or organ dysfunction resulting from the combined LET-SeNPs and X-ray treatment (Figure 7B,C). The overall findings suggest that the therapeutic approach of LET-SeNPs combined with X-ray radiation was well-tolerated by the mice and effectively targeted tumor growth without causing substantial harm to vital organs or systemic functions.

The experimental design for the research involved the use of bilateral tumor-bearing nude mice to mitigate individual differences. During the continuous treatment period of 21 days, significant differences in tumor volume growth curves were observed between the left and right sides of the mice. The growth curve on the left side exhibited a more pronounced increase compared to the right side. Notably, starting from day 9 of treatment, the tumor volume on the left side (no X-ray irradiation group) was significantly greater than that of the right side (combined treatment group), with the difference becoming progressively more significant thereafter. At the end of the 21-day treatment period, the tumor volumes in the left axillary regions of the mice were markedly higher than that on the right sides, with average volumes of 1.17 cm^3^ and 0.66 cm^3^, respectively (Figure 8A,B and Appendix A). After a 28-day treatment period, all tumor specimens underwent examination through H&E staining and immunohistochemical staining for p-p53, as illustrated in Figure 8C. The histological analysis of the tumor specimens revealed promising results. H&E staining showed lower levels of nuclear abnormality and higher tissue differentiation on the side treated with LET-SeNPs combined with X-ray therapy compared to the untreated side. This indicates that the combination treatment led to improved tumor cell morphology and tissue structure, suggesting effective inhibition of tumor growth and progression.

Furthermore, immunohistochemical staining for p-p53 demonstrated a stronger induction of p-p53 expression in the side treated with the combination therapy compared to the untreated side. The upregulation of p-p53, a key tumor suppressor protein involved in cell cycle regulation and DNA repair, further supports the effectiveness of the LET-SeNPs and X-ray combination treatment in inducing cellular responses that inhibit tumor growth.

Overall, the results of the study indicate that the combination of LET-SeNPs and X-ray treatment exhibits strong efficacy against tumors in living organisms while causing minimal harm to the experimental mice. These findings highlight the potential of this therapeutic approach for cancer treatment, emphasizing both its antitumor effects and safety profile in preclinical models.

## 4. Discussion

The current investigation underscores a promising therapeutic approach wherein the co-administration of LET-SeNPs with X-ray irradiation effectively inhibits the proliferation and colony formation of PC3 prostate cancer cells. This combined treatment notably enhances the sensitivity of these cells to X-ray-induced apoptosis, a response attributed to the inhibition of TrxR activity and the subsequent activation of ROS-dependent signaling pathways. Our findings delineate a clear mechanistic framework, whereby the modulation of the p53 and MAPK pathways is implicated in promoting DNA damage and concurrently disrupting cellular DNA repair processes.

In vivo experiments corroborate these findings, revealing that the synergistic effects of LET-SeNPs and X-ray treatment significantly curtail prostate tumor growth, primarily through the upregulation of p53 expression. This highlights a critical role for p53 in the tumor suppression observed with the combination therapy. The innovative integration of LET-SeNPs into the X-ray radiotherapy regimen presents a compelling strategy, demonstrating enhanced anti-prostate cancer efficacy while permitting the administration of lower radiation doses compared to traditional radiotherapy modalities (Figure 9).

Radiotherapy remains a pivotal component in the management of prostate cancer, with a substantial proportion of patients necessitating precise radiation dosages [49]. Preliminary studies have identified LET-SeNPs as potential radiosensitizers, yet the specific mechanisms by which these nanoparticles exert their effects require further elucidation. In response, our study introduced a novel LET-SeNP nanocarrier system, characterized by selenium nanoparticles modified with mushroom polysaccharides, which exhibited marked therapeutic efficacy in conjunction with radiotherapy.

Looking ahead, our research endeavors will focus on refining the design and optimization of this low-toxicity nanocarrier system incorporating selenium nanoparticles, aligned with radiotherapeutic applications. We foresee that this approach could significantly bolster antitumor efficacy and improve patient outcomes. Our findings yield crucial insights that are instrumental for the clinical translation of nanotherapeutics and underscore the precision application of combined modality therapies in oncology. As we move toward implementation in clinical settings, it is imperative to further explore the impact of this combined treatment on tumor dynamics and patient quality of life, thereby cementing its role as a promising radiosensitizer in prostate cancer therapy.

## Figures and Tables

**Figure 1 pharmaceutics-16-01230-f001:**
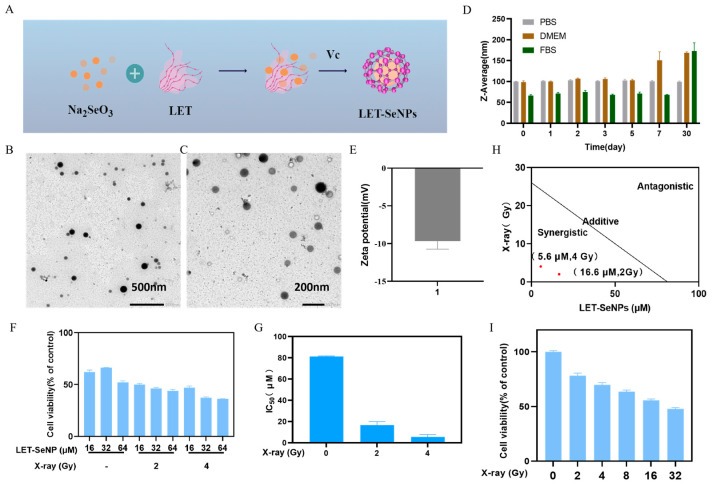
The survival rate of PC3 cells was studied by examining the effects of various levels of LET-SeNPs and doses of X-ray. (**A**) Process flow diagram of LET-SeNP preparation. (**B**,**C**) TEM image of LET-SeNPs. (**D**) Stability of LET-SeNPs in PBS, DMEM, and FBS. (**E**) Zeta potential of LET-SeNPs. (**F**) Cytotoxicity of LET-SeNPs (16, 32, and 64 μM) with X-ray (0, 2, and 4 Gy) on PC3 cells. (**G**) PC3 cells were exposed to X-ray (0, 2, and 4 Gy) irradiation followed by incubation for 72 h. (**H**) LET-SeNPs combined with X-ray (2 and 4 Gy) enhances the anticancer efficacy. An isobologram was used to analyze the combined effect of X-ray (2 and 4 Gy) radiation and LET-SeNPs (16.6 μM, 5.6 μM) on inhibiting the growth of PC3 cells. (**I**) The viability of PC3 cells was assessed after exposure to X-ray (0, 2, 4,8, 16, and 32 Gy). Each value represents means ± SD (n = 3).

**Figure 2 pharmaceutics-16-01230-f002:**
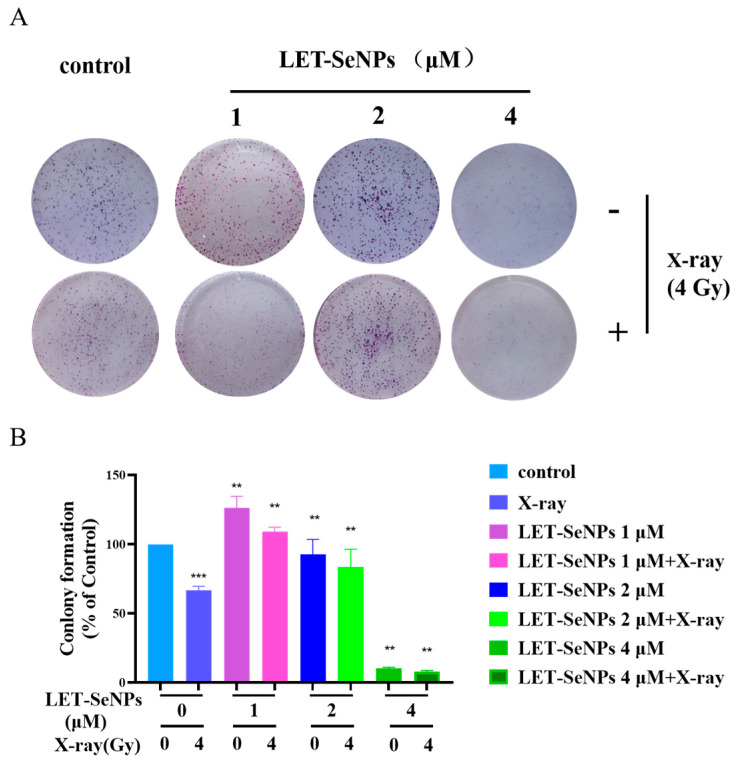
The combination of LET-SeNPs and X-ray (4 Gy) suppressed the clonogenic ability of PC3 cells. (**A**) The combination of various levels of LET-SeNPs (1, 2, and 4 μM) and X-ray (4 Gy) affected the ability of PC3 cells to form colonies and survive. (**B**) Colony assay of PC3 cells was assessed when treated with varying doses of LET-SeNPs (1, 2, and 4 μM) in combination with X-ray (4 Gy) radiation. Each value represents mean ± SD (n = 3). ** *p* < 0.01, *** *p* < 0.001.

**Figure 3 pharmaceutics-16-01230-f003:**
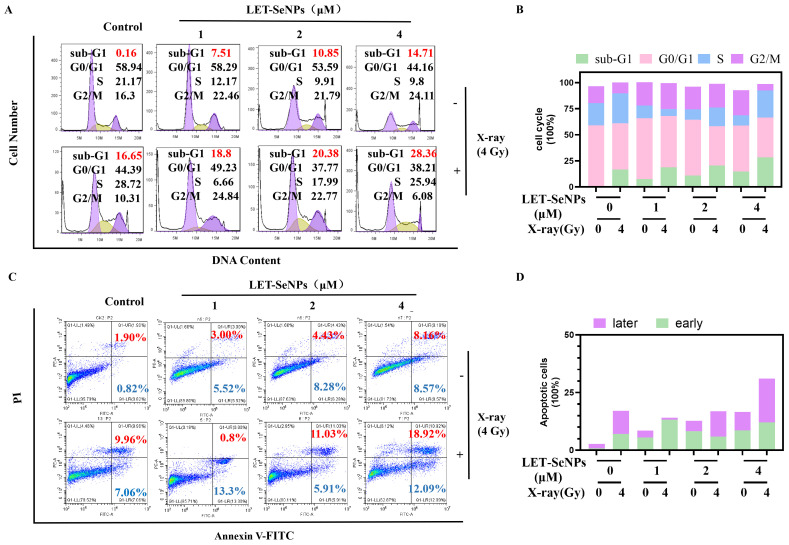
Cell cycle arrest and apoptosis analysis after treatment with LET-SeNPs combined with X-ray (4 Gy) radiotherapy. (**A**) Flow cytometric analysis of PC3 cells under treatment with or without different concentrations of LET-SeNPs (1, 2, and 4 μM) and X-ray (4 Gy). (**C**) Flow cytometry apoptosis of PC3 cells treatment with or without different concentrations of LET-SeNPs (1, 2, and 4 μM) and X-ray (4 Gy). (**B**,**D**) Quantitative analysis of PC3 cell cycle arrest and apoptosis.

**Figure 4 pharmaceutics-16-01230-f004:**
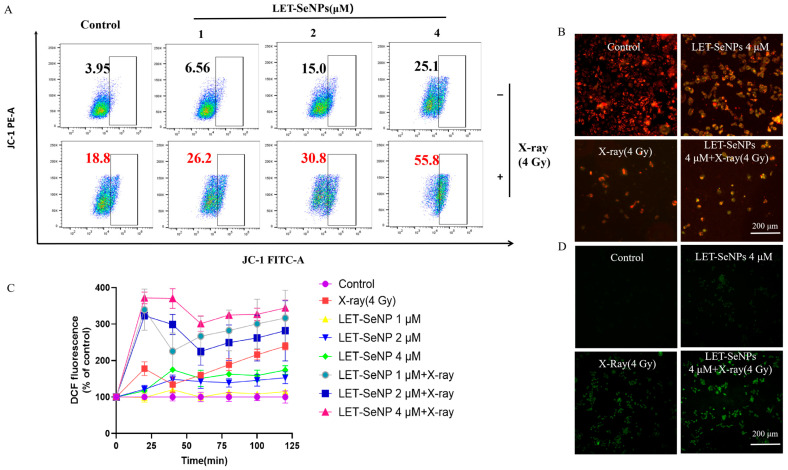
After 2 h of treatment with LET-SeNPs combined with X-ray radiotherapy (4 Gy), the mitochondrial membrane potential and ROS levels in PC3 cells were measured. (**A**) Flow cytometric analysis of the mitochondrial membrane potential of PC3 cells after treatment with various concentrations of LET-SeNPs (1, 2, and 4 μM) and X-ray (4 Gy). (**B**) JC-1 fluorescence detected by fluorescent microscope after treatment with varying concentrations of LET-SeNPs (4 μM) and X-ray (4 Gy). (**C**) ROS levels of PC3 cells treated by LET-SeNPs (1, 2, and 4 μM) and X-ray (4 Gy). (**D**) Fluorescence imaging in PC3 cells marked with DCF.

**Figure 5 pharmaceutics-16-01230-f005:**
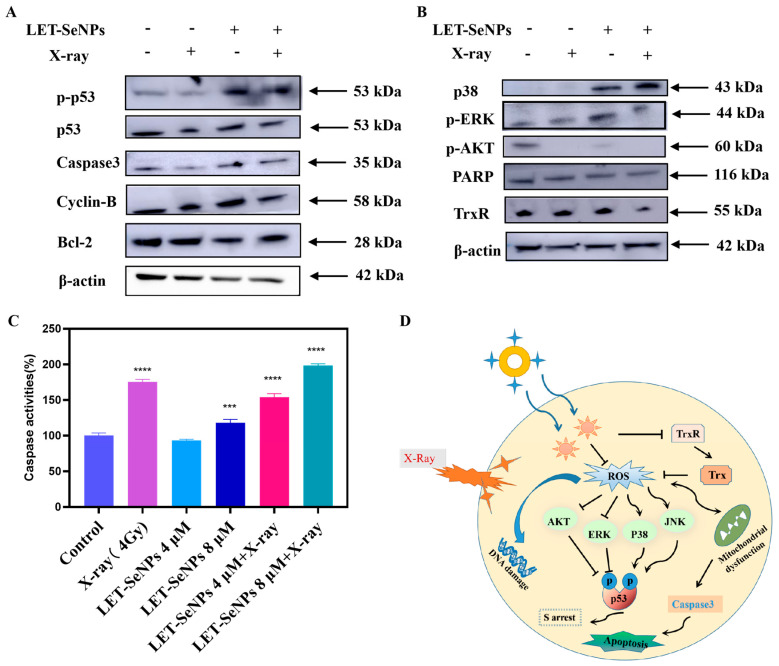
Activation of apoptotic signaling pathways by LET-SeNPs and X-ray (4 Gy). (**A**) LET-SeNPs combined with X-ray (4 Gy) enhanced p53 signal pathway. (**B**) LET-SeNPs combined with X-ray (4 Gy) effects the expression of MAPKs and AKT pathways in PC3 cells, and the expression levels of TrxR and PARP. (**C**) Activation of caspase-3 pathway in different groups. *** *p <* 0.001, **** *p* < 0.0001. (**D**) LET-SeNPs combined with X-ray (4 Gy) proposed signal pathways of apoptosis.

**Figure 6 pharmaceutics-16-01230-f006:**
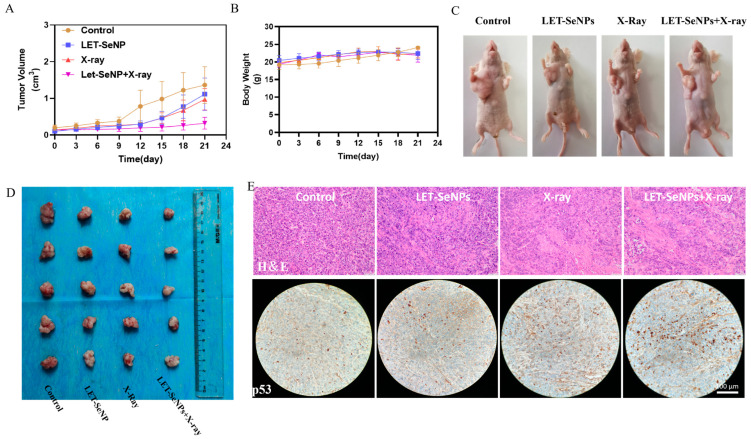
In vivo anticancer effects of LET-SeNPs and X-ray tested in nude mice. (**A**,**B**) The growth curves and body weight of tumors on mice treated with LET-SeNPs and X-ray. (**C**,**D**) The images of representative mice and tumors in each group. (**E**) Microscopic pictures of tumor sections stained with H&E and immunohistochemical p-p53 from various groups.

**Figure 7 pharmaceutics-16-01230-f007:**
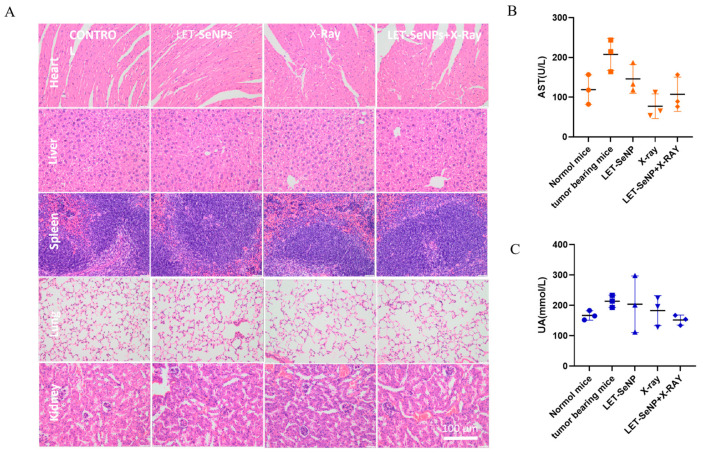
Major organs and blood biochemical indexes were evaluated. (**A**) Heart, liver, spleen, lung, and kidney in 4 groups were assessed for potential toxicity by H&E staining. (**B**,**C**) AST and UA were analyzed to evaluate systemic toxicity.

**Figure 8 pharmaceutics-16-01230-f008:**
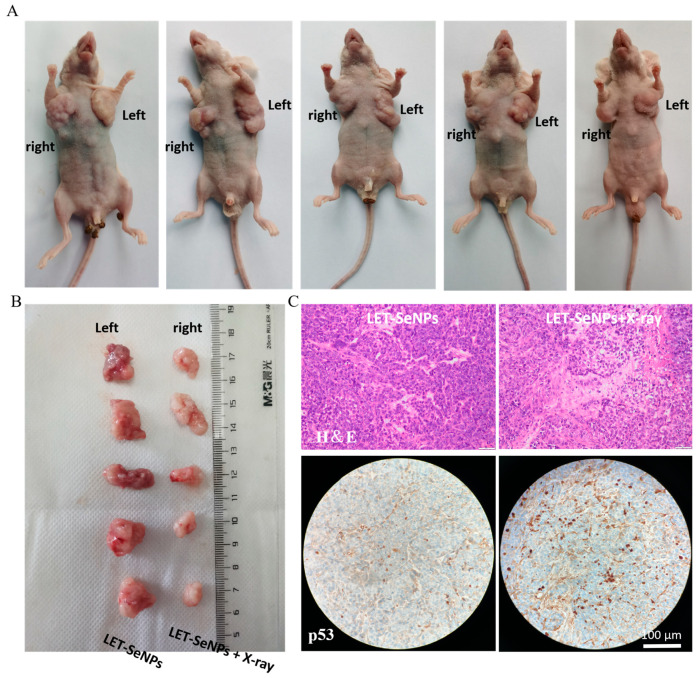
Schematic diagram of bilateral tumor-bearing mouse model. (**A**,**B**) After 21 days of treatment with LET-SeNPs and X-ray (4 Gy), images of mice and tumors were taken. The left sides of the mice were treated with LET-SeNPs (2 mg/kg), while the right sides were treated with LET-SeNPs (2 mg/kg) in combination with X-ray (4 Gy). (**C**) Microscope images of H&E and p-p53 immunohistochemical staining of tumors from nude mice with tumors on both sides.

**Figure 9 pharmaceutics-16-01230-f009:**
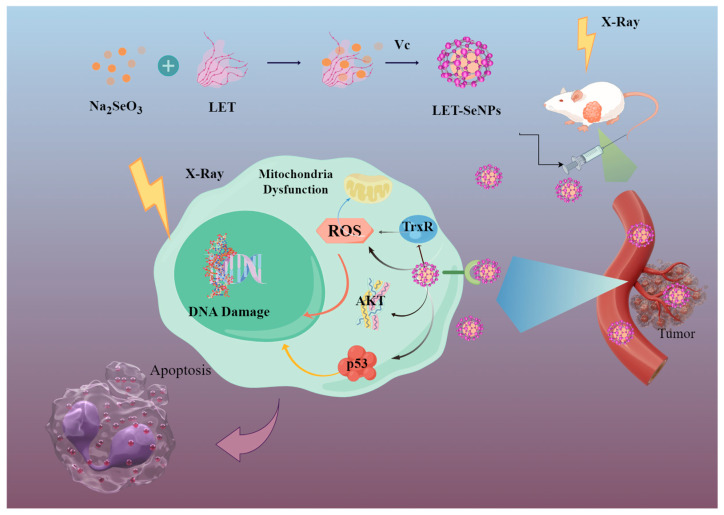
The proposed approach includes the utilization of functionalized LET-SeNPs to increase the efficacy of radiotherapy on prostate cancer PC3 cells. The joint therapy efficiently induces excessive ROS production within cells to control the p53-related DNA damage apoptosis signaling pathway and phosphorylation of MAPK and AKT. (By Figdraw, www.figdraw.com).

## Data Availability

No new data were created or analyzed in this study.

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
