# Peer review of "Enhancing Radiotherapy Sensitivity in Prostate Cancer with Lentinan-Functionalized Selenium Nanoparticles: Mechanistic Insights and Therapeutic Potential"

_pharmaceutics, 2024, doi:10.3390/pharmaceutics16091230_

Round 1

Reviewer 1 Report

Comments and Suggestions for Authors

Present article describe the use of Lentinan-functionalized Selenium Nanoparticles as an anti-tumour agent enhancing the radiotherapy efficacy.

The introduction provides enough information about the interest of the use of selenium materials as prevention and treatment in cancer, and the number of cites is appropiate. The methodology is well described and that section contains enough information to reproduce the experiments. The experiments were well designed and the concluions are supported by the data obtained. The article gives a novel approaching to a high interest research area to the audience of Pharmaceutics.

I recommend accept in present form.

Author Response

The introduction provides enough information about the interest of the use of selenium materials as prevention and treatment in cancer, and the number of cites is appropiate. The methodology is well described and that section contains enough information to reproduce the experiments. The experiments were well designed and the concluions are supported by the data obtained. The article gives a novel approaching to a high interest research area to the audience of Pharmaceutics.

Reply: Thank you for your comments. In the preliminary phase of this study, extensive preparatory work was conducted to ensure the smooth progression of the experiments. Each experiment underwent rigorous scrutiny, with a high standard set for the accuracy of the experimental data. Every experiment was repeated at least three times to obtain reliable data support. We sincerely appreciate your recognition of this research.

Reviewer 2 Report

Comments and Suggestions for Authors

The manuscript entitled "Enhancing Radiotherapy Sensitivity in Prostate Cancer with Lentinan-functionalized Selenium Nanoparticles: Mechanistic Insights and Therapeutic Potential" raises the important issue of increasing the effectiveness of cancer treatment. The authors conducted valuable research, but it would be worthwhile to supplement the manuscript with a few issues.

The manuscript can be improved according to the following suggestions:

1) The authors chose the shortcut LET for lentinan, for example: Lentinan-functionalized Selenium Nanoparticles (LET-SeNPs). In the literature relating to radiobiology and radiation effects, the abbreviation LET is usually reserved for "Linear Energy Transfer". In the literature lentinan is usually abbreviated as LNT (however, this shortcut is also not perfect, because LNT is usually associated with "linear no-threshold".). Maybe the authors have an idea for a better shortcut?

2) All abbreviations and acronyms should be explained when they first appear in the text. For example in the abstract: MAPK - (mitogen-activated protein kinase), TrxR (thioredoxin reductase), etc.

3) In the introduction the authors mentioned that effective radiosensitizers are needed for enhancing radiotherapy outcomes. It is known from the literature (see for example review paper: www.researchgate.net/publication/317588178_Nanoparticles_as_radiosensitizers_in_photon_and_hadron_radiotherapy ) that in case of nanoparticles the radiosensitizing effect is mainly related to the intensifying of the emission of photoelectrons and Auger electrons after the X-ray irradiation. The electrons emitted as a result of photon interaction with nanoparticles are responsible for the increased generation of reactive oxygen species. In general, radiobiological and biochemical effects originate from fundamental physical interactions. Selenium particles have a smaller atomic number and different photoelectric cross section at kilovoltage energy in comparison to Au, Gd and Pt nanoparticles. It will be good if the authors at least mention in the introduction the fundamental physical processes responsible for sensitization and refer to the available literature.

4) In the "Materials and chemistry" subsection the authors wrote "DCF probe, (JC-1) was purchased from Sigma". It is better to write the full names of the companies along with their headquarters, e.g.  dichlorofluorescein (DCF) from Sigma-Aldrich (Massachusetts, US).

5) The fonts of the captions (e.g. apoptosis) in the scheme 1 could be larger and therefore more legible on an A4 printout.

6) The "Materials and Methods" section lacks information about the irradiation process (type of accelerator, photon energy, dose rate, etc.). Energy and dose rate are crucial when assessing the effects of radiation exposure.

7) Was a homogeneous distribution of nanoparticles achieved in the region of cancer cells in the mouse model? How was this possible after injecting nanoparticles into the bloodstream? How was the dose (2 mg/kg) selected? How the opsonization of nanoparticles was avoided? It would be good if the authors addressed these issues in the manuscript.

8) Panel c in Figure 3 is poorly legible on A4 printout.

9) Is it known what kind of reactive oxygen species were produced during treatment with LET-SeNPs combined with X-ray radiotherapy? Although it is difficult to assess this without using EPR spectroscopy, do the authors have any suggestions?

Comments on the Quality of English Language

The language of the manuscript is generally understandable and the work does not contain many glaring linguistic errors.

Author Response

1) The authors chose the shortcut LET for lentinan, for example: Lentinan-functionalized Selenium Nanoparticles (LET-SeNPs). In the literature relating to radiobiology and radiation effects, the abbreviation LET is usually reserved for "Linear Energy Transfer". In the literature lentinan is usually abbreviated as LNT (however, this shortcut is also not perfect, because LNT is usually associated with "linear no-threshold".). Maybe the authors have an idea for a better shortcut?

Reply: Thank you for your comments. Your suggestions have greatly contributed to the precision of the article's expression. Our laboratory has conducted extensive preliminary research on the functionality of LET-SeNPs, and our recently published articles consistently use the abbreviation LET-SeNPs. The alternative abbreviation you proposed, LNT, has also been reported in the literature. However, we have chosen to adopt the commonly used abbreviation within our laboratory for this manuscript and hope to receive your understanding in this regard.

The relevant literature is as follows:[1] S. Liu, N. Li, H. Lai, L. Xu, Y. Zeng, X. Chen, H. Huang, T. Chen, J. Liu, J. Wang, Selenium Nanoparticles Enhance NK Cell-Mediated Tumoricidal Activity in Malignant Pleural Effusion via the TrxR1-IL-18RAP-pSTAT3 Pathway. Adv. Funct. Mater. 2024, 34, 2401264.

[2] Nie S, He X, Sun Z, Zhang Y, Liu T, Chen T, Zhao J. Selenium speciation-dependent cancer radiosensitization by induction of G2/M cell cycle arrest and apoptosis. Front Bioeng Biotechnol. 2023 Mar 22;11:1168827. doi: 10.3389/fbioe.2023.1168827. PMID: 37034255; PMCID: PMC10073679.

2) All abbreviations and acronyms should be explained when they first appear in the text. For example in the abstract: MAPK - (mitogen-activated protein kinase), TrxR (thioredoxin reductase), etc.

Reply: Thank you for your comments. Thank you for your suggested revisions, which have further enhanced the accuracy of the article. Corrections have been made accordingly in lines 28, 30, and 33 on page 2.

3) In the introduction the authors mentioned that effective radiosensitizers are needed for enhancing radiotherapy outcomes. It is known from the literature (see for example review paper: www.researchgate.net/publication/317588178_Nanoparticles_as_radiosensitizers_in_photon_and_hadron_radiotherapy) that in case of nanoparticles the radiosensitizing effect is mainly related to the intensifying of the emission of photoelectrons and Auger electrons after the X-ray irradiation. The electrons emitted as a result of photon interaction with nanoparticles are responsible for the increased generation of reactive oxygen species. In general, radiobiological and biochemical effects originate from fundamental physical interactions. Selenium particles have a smaller atomic number and different photoelectric cross section at kilovoltage energy in comparison to Au, Gd and Pt nanoparticles. It will be good if the authors at least mention in the introduction the fundamental physical processes responsible for sensitization and refer to the available literature.

Reply: Thank you for your comments. Radiation sensitizers generally enhance the efficacy of radiotherapy through three primary mechanisms:

  1. Improving Hypoxia
    Rapid tumor growth often leads to a hypoxic environment, which negatively impacts the effectiveness of radiotherapy. Research indicates that the radiation dose required to eradicate tumor cells under hypoxic conditions is 2-3 times higher than that required under aerobic conditions. Therefore, improving hypoxia can enhance the effectiveness of radiotherapy.
  2. Regulating the Cell Cycle
    The sensitivity of tumor cells to radiotherapy is dependent on cell cycle regulation. Radiotherapy activates checkpoints for DNA damage, with G2-M phase cells being most sensitive to radiation, while G1 phase cells exhibit lower sensitivity. If cell cycle arrest occurs during the G2-M phase, the sensitivity of the tumor to radiotherapy increases.
  3. Promoting DNA Damage
    The key to radiotherapy is inducing DNA damage in tumor cells, primarily through pathways such as single and double-strand breaks as well as base damage. Double-strand breaks are the predominant form of damage, and nanodrugs can inhibit DNA repair mechanisms, thereby enhancing the effectiveness of radiotherapy.

References are as follows: [1] MICHON S, RODIER F, YU F T H. Targeted Anti-Cancer Provascular Therapy Using Ultrasound, Microbubbles, and Nitrite to Increase Radiotherapy Efficacy [J]. Bioconjug Chem, 2022, 33(6): 1093-105.

[2] HU X, HU J, PANG Y, et al. Application of nano-radiosensitizers in non-small cell lung cancer [J]. Front Oncol, 2024, 14(1372780.

[3] ZHOU J, LEI N, QIN B, et al. Aldolase A promotes cervical cancer cell radioresistance by regulating the glycolysis and DNA damage after irradiation [J]. Cancer Biol Ther, 2023, 24(1): 2287128.

[4] SERBAN R M, NICULAE D, MANDA G, et al. Modifications in cellular viability, DNA damage and stress responses inflicted in cancer cells by copper-64 ions [J]. Front Med (Lausanne), 2023, 10(1197846.

[5] REUVERS T G A, VERKAIK N S, STUURMAN D, et al. DNA-PKcs inhibitors sensitize neuroendocrine tumor cells to peptide receptor radionuclide therapy in vitro and in vivo [J]. Theranostics, 2023, 13(10): 3117-30.

The corresponding modifications can be found on pages 4-5 of the article, lines 66-79.

The combination of LET-SeNPs with X-ray therapy has been found to enhance reactive oxygen species build-up in cancer cells, leading to DNA damage mediated by mitochondria。This is mentioned in lines 77-79 on page 5, along with the corresponding references.

4) In the "Materials and chemistry" subsection the authors wrote "DCF probe, (JC-1) was purchased from Sigma". It is better to write the full names of the companies along with their headquarters, e.g. dichlorofluorescein (DCF) from Sigma-Aldrich (Massachusetts, US).

Reply: Thank you for your comments. I have made the modifications based on your suggested improvements in lines 101-103 on page 6.

5) The fonts of the captions (e.g. apoptosis) in the scheme 1 could be larger and therefore more legible on an A4 printout.

Reply: Thank you for your comments. The scheme 1 has been optimized.

6) The "Materials and Methods" section lacks information about the irradiation process (type of accelerator, photon energy, dose rate, etc.). Energy and dose rate are crucial when assessing the effects of radiation exposure.

Reply: Thank you for your comments. Your inquiries regarding the specific parameters in radiotherapy have added rigor to the article. A clarification has been provided in lines 187-189 on page 10. All radiation experiments are conducted using an Elekta Precise linear accelerator, which operates at 6 MeV energy, provides a penetration depth of 5 cm, and delivers 100 monitor units (MU) in a hospital setting.

7) Was a homogeneous distribution of nanoparticles achieved in the region of cancer cells in the mouse model? How was this possible after injecting nanoparticles into the bloodstream? How was the dose (2 mg/kg) selected? How the opsonization of nanoparticles was avoided? It would be good if the authors addressed these issues in the manuscript.

Reply: Thank you for your comments.

The LET-SeNPs selected for this study are based on extensive preliminary experiments. Due to various factors, this experiment did not include optical imaging of the nanoparticles in vivo, and therefore, we cannot obtain information on the targeting accumulation of the particles. However, previous research has concluded that SeNPs modified with carbohydrate substances can rapidly enter the lysosomes of tumor cells through endocytosis and subsequently diffuse into the cytoplasm.

References are as follows: Wu H , Li X , Liu W ,et al.Surface decoration of selenium nanoparticles by mushroom polysaccharides–protein complexes to achieve enhanced cellular uptake and antiproliferative activity[J].Journal of Materials Chemistry, 2012, 22.DOI:10.1039/c2jm16828f.

Based on a summary of extensive preliminary animal experiments conducted in the laboratory, we also screened for tail vein injection and direct tumor injection. Ultimately, we chose the tail vein injection method due to its simplicity and better reproducibility. Generally, the dosages selected for tail vein injection were 1, 1.5, 2, and 2.5 mg/kg, and we finalized the middle value to ensure the reproducibility of the experiment.

References are as follows: Zeng, D.; Zhao, J.; Luk, K. H.; Cheung, S. T.; Wong, K. H.; Chen, T. Potentiation of in Vivo Anticancer Efficacy of Selenium Nanoparticles by Mushroom Polysaccharides Surface Decoration. J Agric Food Chem 2019, 67 (10), 2865-2876. DOI: 10.1021/acs.jafc.9b00193.

8) Panel c in Figure 3 is poorly legible on A4 printout.

Reply: Thank you for your comments. Figure 3c has been optimized.

9) Is it known what kind of reactive oxygen species were produced during treatment with LET-SeNPs combined with X-ray radiotherapy? Although it is difficult to assess this without using EPR spectroscopy, do the authors have any suggestions?

Reply: Thank you for your comments. Currently, the experiments in our research group regarding the measurement of ROS levels with LET-SeNPs combined with radiotherapy primarily utilize the DCF probe for detection, which measures total oxygen content and cannot distinguish between superoxide anions (O2−), hydrogen peroxide (H2O2), hydroxyl radicals (HO-), or singlet oxygen (1O2). However, for other compounds of SeNPs, our group has conducted further analysis on the levels of superoxide anions and singlet oxygen using DHE and DPBF probes to evaluate their synergistic effects.

References are as follows:

[1] Leung, Chan,Lizhen, He,Binwei, Zhou et al. Cancer-Targeted Selenium Nanoparticles Sensitize Cancer Cells to Continuous γ Radiation to Achieve Synergetic Chemo-Radiotherapy.[J] .Chem Asian J, 2017, 12: 0.

[2] Zhenhuan, Song,Ting, Liu,Haoqiang, Lai et al. A Universally EDTA-Assisted Synthesis of Polytypic Bismuth Telluride Nanoplates with a Size-Dependent Enhancement of Tumor Radiosensitivity and Metabolism In Vivo.[J] .ACS Nano, 2022, 16: 0.

Reviewer 3 Report

Comments and Suggestions for Authors

The MS submitted by Zou et al, titled Enhnacing Radiotherapy Sensitivity in Prostate Cancer with Lentinan-functional Selenium Nanoparticles: Mechanistic Insights and Therapeutic Potenial is very thoughtful exploration of study. However I have few suggestions for Authors.

1) Scheme1 should me in word digram, If authors want to put this pic as ascheme please level this as figure 1 and chnage the entire MS accordingly.

2) Introduction is well written, but it seems few information uncessary written. Authors should shorten the Intro part accordingly.

3) Why the TEM image ( figure 1 BC) having different scale bar (500 nm & 200 nm). Authors should provide explanation for this.

4) Figure 3 needs little bit more resolution.

5) Figure 4 (B &D). Authors should put scale bar in each figure. Without scale bar this results not publication garde.

6) Where is dicussion ?

7) Conclusion is so long.

8) Add limitation of the study.

Author Response

1) Scheme1 should me in word digram, If authors want to put this pic as ascheme please level this as figure 1 and chnage the entire MS accordingly.

Reply: Thank you for your comments. Scheme 1 is a TOC (Table of Contents) figure that I created personally, encompassing the principles and highlights of my research. I aim for it to be simple and easy to read, serving as a starting point for further discussion.

2) Introduction is well written, but it seems few information uncessary written. Authors should shorten the Intro part accordingly.

Reply: Thank you for your comments. The Introduction of the article aims to provide a detailed overview to facilitate understanding. However, if the journal has specific length requirements, we can make appropriate modifications and edits as necessary.

3) Why the TEM image ( figure 1 BC) having different scale bar (500 nm & 200 nm). Authors should provide explanation for this.

Reply: Thank you for your comments. In the experiments, images are typically captured at different magnifications to enable subsequent selection. This is particularly relevant for TEM imaging, where the scale bar varies depending on the chosen magnification.

4) Figure 3 needs little bit more resolution.

Reply: Thank you for your comments. The unclear areas in Figure 3 have been processed to achieve the best publication quality.

5) Figure 4 (B &D). Authors should put scale bar in each figure. Without scale bar this results not publication garde.

Reply: Thank you for your comments. Figure 4 (B &D) has been optimized.

6) Where is dicussion ?

Reply: Thank you for your comments. The third chapter of this article is titled "RESULTS and DISCUSSION," so a separate section for discussion has not been allocated. I will make revisions based on your suggestions in lines 497-509 on pages 27-28 of the article.

7) Conclusion is so long.

Reply: Thank you for your comments. Based on your feedback, the Conclusion has been rewritten to include summaries, limitations, and future perspectives.

8) Add limitation of the study

Reply: Thank you for your comments. The limitations have been added to the Conclusion, specifically in lines 510-523 on pages 28-29 of the article.

Round 2

Reviewer 3 Report

Comments and Suggestions for Authors

It seems authors are not paying proper attention to the comments.

1) Scheme 1 , authors claimed that this is TOC (table of content). This is not table , its a proper digram. Authors should revised accordingly.

2) Results & discussion both together. Authors should understand this is research article not a book chapter. Authors should check the scientific article first and revise the MS accordingly.

Other responses are accepatble but without addressing these 2 , MS not publication grade. 

Author Response

Comments 1: Scheme 1, authors claimed that this is TOC (table of content). This is not table, its a proper digram. Authors should revised accordingly.

Reply: Thank you for your comments. Thank you for your suggested revisions, which have further enhanced the accuracy of the article. I have made the modifications based on your suggested improvements in lines 487-492 on page 28.

Comments 2: Results & discussion both together. Authors should understand this is research article not a book chapter. Authors should check the scientific article first and revise the MS accordingly.

Other responses are accepatble but without addressing these 2 , MS not publication grade.

Reply: Thank you very much for your valuable review comments. Your feedback has been instrumental in enhancing the accuracy and readability of the manuscript. In accordance with your suggestions, we have made detailed revisions to the text, specifically in lines 494-527 on pages 29-30. Additionally, we have adjusted the formatting of the manuscript to meet publication standards. We sincerely appreciate your insightful input.

Round 3

Reviewer 3 Report

Comments and Suggestions for Authors

Authors revise the MS as suggested. MS quality improved now. I recommend this MS for publication.